# Acetyl-DL-leucine in two individuals with REM sleep behavior disorder improves symptoms, reverses loss of striatal dopamine-transporter binding and stabilizes pathological metabolic brain pattern—case reports

Wolfgang H. Oertel [1,2] ✉, Annette Janzen[1], Martin T. Henrich[1,3,4], Fanni F. Geibl[1,3,4], Elisabeth Sittig[1], Sanne K. Meles[5], Giulia Carli[6], Klaus Leenders[6], Jan Booij[7], D. James Surmeier [4], Lars Timmermann[1] & Michael Strupp [8] ✉

Isolated REM Sleep Behavior Disorder (iRBD) is considered a prodrome of Parkinson's disease (PD). We investigate whether the potentially disease-modifying compound acetyl-DL-leucine (ADLL; 5 g/d) has an effect on pro-dromal PD progression in 2 iRBD-patients. Outcome parameters are RBD-severity sum-score (RBD-SS-3), dopamine-transporter single-photon emission computerized tomography (DAT-SPECT) and metabolic "Parkinson-Disease-related-Pattern (PDRP)"-z-score in $^{18}$F-fluorodeoxyglucose positron emission tomography (FDG-PET). After 3 weeks ADLL-treatment, the RBD-SS-3 drops markedly in both patients and remains reduced for >18 months of ADLL-treatment. In patient 1 (female), the DAT-SPECT putaminal binding ratio (PBR) decreases in the 5 years pretreatment from normal (1.88) to pathological (1.22) and the patient's FDG-PET-PDRP-z-score rises from 1.72 to 3.28 (pathological). After 22 months of ADLL-treatment, the DAT-SPECT-PBR increases to 1.67 and the FDG-PET-PDRP-z-score stabilizes at 3.18. Similar results are seen in patient 2 (male): his DAT-SPECT-PBR rises from a pre-treatment value of 1.42 to 1.72 (close to normal) and the FDG-PET-PDRP-z-score decreases from 1.02 to 0.30 after 18 months of ADLL-treatment. These results support exploration of whether ADLL may have disease-modifying properties in prodromal PD.

Isolated REM Sleep Behavior Disorder (iRBD) is a specific and common prodromal phenotype of Parkinson's Disease (PD). About 25% of patients with early PD report having experienced aggressive dream content and dream enactment—the two key symptoms of RBD—before the manifestation of motor Parkinson's symptoms[1,2]. Furthermore, iRBD patients carry a >85% risk of a conversion to PD or its variant Dementia with Lewy Bodies (DLB) within 10–15 years[3,4]. Slowing, stopping or even

reversing the progression of PD in the prodromal stage would be a major advance for human health. Currently, however, there are no proven therapeutic strategies for slowing down the progression of iRBD or PD.

In 2020, we treated a PD patient, who also suffered from Restless Legs Syndrome (RLS) and the phenotype RBD, with the modified amino acid acetyl-D-leucine (ADLL; 5 g/day orally). Within 2 weeks after initiating ADLL treatment, the patient's RLS symptoms had

---

substantially abated[5]. In addition, after 5 weeks of ADLL therapy, the patient reported an improvement in the two key clinical features of iRBD: the disappearance of aggressive dreams and a considerable reduction of dream enactment.

Based on this observation, we selected two patients with iRBD and offered them treatment with ADLL. The effect of ADLL on the severity of the RBD phenotype was "clinically" recorded daily by patient and spouse by means of an RBD-diary. For objective monitoring, the effect of ADLL on the neurodegeneration of the dopaminergic nigrostriatal pathway was investigated with dopamine-transporter single-photon emission computerized tomography (DAT-SPECT). For studying the effect of ADLL on the pathological metabolic "Parkinson-Disease-related-Pattern (PDRP)"-z-score, we employed [18]F-fluorodeoxyglucose positron emission tomography (FDG-PET).

Acetyl-leucine (AL) has been found to have symptomatic and disease-modifying effects in animal models of lysosomal storage disorders (LSD), including Niemann-Pick disease type C (NPC) and GM2 gangliosidosis[6,7]. Several formal LSD clinical trials with the active L-enantiomer, including our recent double-blind, placebo-controlled crossover phase 3 trial in NPC[8], found that N-acetyl-L-leucine had rapid beneficial effects on neurological signs and symptoms and an excellent safety profile[8–10]. The agent enters enzyme-controlled pathways that correct metabolic dysfunction and improves energy adenosine triphosphate (ATP) production[6,7]. Lysosomal and mitochondrial dysfunctions have been proposed as important factors in the pathogenesis of PD[11–14]. Therefore, AL might also have a favorable impact on the prodromal stage of PD by slowing down its progression already in the stage of iRBD.

The results show that ADLL therapy has effects on three outcome parameters in the 2 iRBD patients: it markedly improves the severity of the RBD phenotype (reduction of aggressive dream content and dream enactment), reverses the loss of striatal dopamine-transporter binding in the nigrostriatal system, and stabilizes the PD-typical pathological metabolic brain pattern. According to the data, it appears—in principle—to be possible to stop, if not reverse the progression of Parkinson's disease in the prodromal iRBD-stage.

## Results

### RBD severity sum-score and clinical evaluation with motor, cognitive, autonomic, and olfactory function tests

Patient 1 (female) had a subjective 3-week RBD-severity sum-score (RBD-SS-3) of 21 before a treatment effect was observed (Fig. 1a). At baseline, olfactory function tests revealed anosmia with a Threshold-Discrimination-Identification (TDI)-sum-score of 6. ADLL treatment was started in November 2021. Three weeks later, the patient's RBD-SS-3 decreased to 5 (Fig. 1a and Supplementary Fig. 1a) and aggressive dream content was nearly absent (Supplementary Table 2). This effect remained sustained at this low level for the following 22 months of continuous ADLL treatment.

Patient 2 (male) had a subjective 3-week RBD-severity sum-score (RBD-SS-3) of 7 before a treatment effect was observed, and thus was less severely affected by RBD symptoms than patient 1 (Fig. 1b). Olfactory function tests revealed anosmia with a TDI-sum-score of 10 at baseline. ADLL therapy was started in January 2022. Three weeks later, the RBD-SS-3 decreased from 7 to 0. The aggressive content of his dreams disappeared entirely (Fig. 1b, Supplementary Fig. 1b, and Supplementary Table 2). These improvements were—apart from an occasional re-appearance of dream enactment due to the use of alcohol (see Section C in Supplementary Information)—maintained for the next 18 months of continuous ADLL treatment.

At the end of the 18 to 22-months, ADLL treatment period patient and spouse were asked to answer questions related to the symptoms

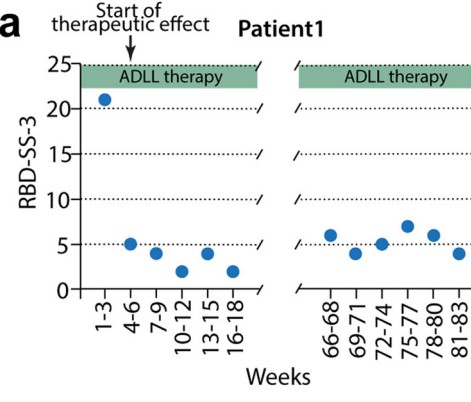

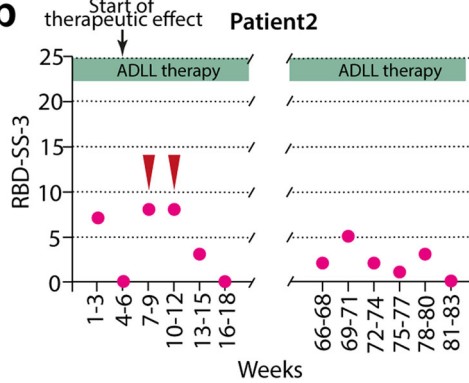

**Fig. 1 | The time course of the severity of the REM sleep behavior disorder phenotype assessed by means of an RBD-diary and expressed as the 3-week RBD-severity sum-score (RBD-SS-3) in patients 1 and 2.** The circles depict the sum of 21 subsequent daily (3 weeks) subjectively recorded severity scores of the RBD-phenotype of patients 1 (**a**) and 2 (**b**). The x-axis covers the first 18 weeks (left panel) and the last 18 weeks (right panel) of the 83-week therapy with ADLL. The red arrows in (**b**) indicate the consumption of alcohol restricted to a total of 5 days during weeks 7–9 and weeks 10–12 by patient 2 leading to an aggravation of the RBD severity. Following week 12, patient 2 stopped the consumption of alcohol (see Section C in Supplementary Information). Source data are provided as a Source Data file.

of the RBD-phenotype. The responses are detailed in Supplementary Table 2.

In Fig. 2, the temporal relationship between the time points of the clinical assessments (motor function test: UPDRS part III or MDS-UPDRS part III; cognitive function test: MOCA; olfactory function test: TDI sum-score) and the time points of the imaging procedures (DAT-SPECT, FDG-PET) are illustrated.

In both patients, regularly assessed UPDRS III/MDS-UPDRS III (motor score—without and with the item "action tremor") and SCOPA-AUT scores remained normal during ADLL therapy (Fig. 2a, b and Supplementary Table 1). Results of the cognitive screening test MoCA stayed in the normal range for patient 2, whereas patient 1 showed a trend of decrease in the MoCA score and developed a mild cognitive impairment during the study. In respect to the exploration of the olfactory function, the TDI sum-score slightly increased over time with ADLL therapy, but remained still in the anosmic range (see also Section D in Supplementary Information). Overall, based on the assessment of the responsible neurologist (AJ) no phenoconversion to Parkinson's disease or dementia with Lewy bodies was observed in both patients during the treatment period with ADLL over 18–22 months. Both patients had a reduced cardiac sympathetic innervation as demonstrated by [123]I-metaiodobenzylguanidine scintigraphy (MIBG).

No adverse effects were reported in either patient with ADLL treatment.

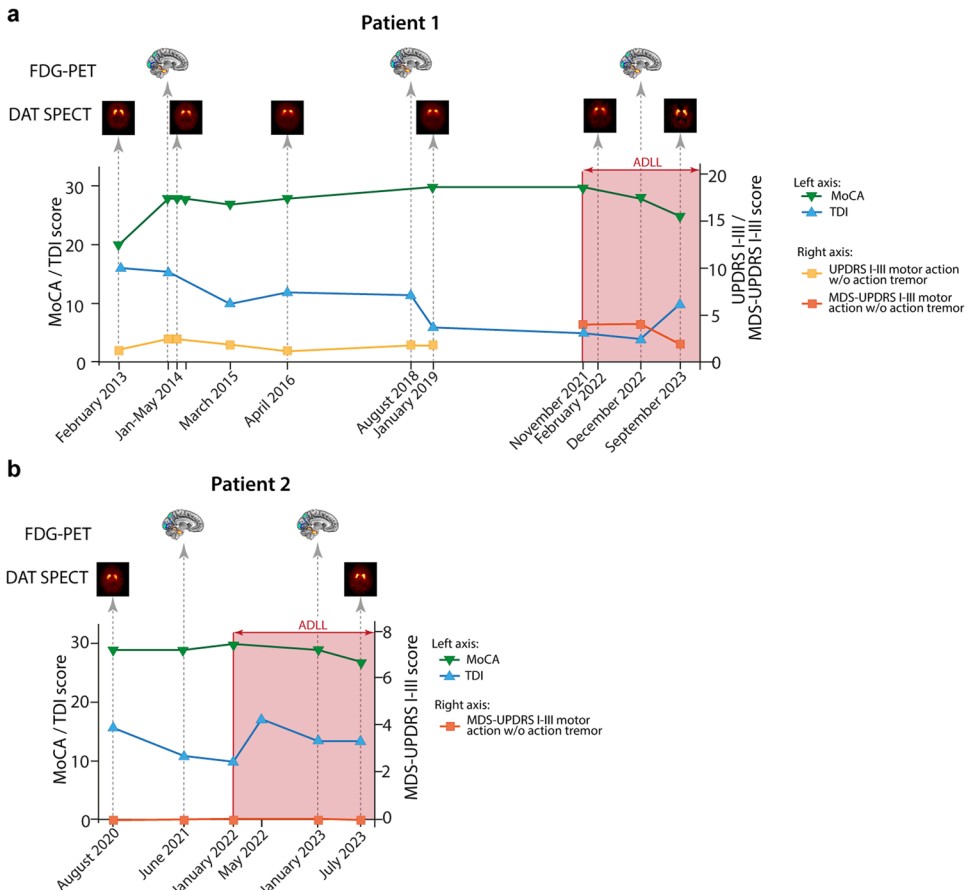

**Fig. 2 | Clinical evaluations and imaging procedures of the patients over time.** The temporal relationship between the time points of the clinical assessments (motor function test: UPDRS part III or MDS-UPDRS part III; cognitive function test: MOCA; olfactory function test: TDI sum-score) and the time points of the imaging procedures (DAT-SPECT, FDG-PET) is illustrated. The red-shaded area indicates therapy with Acetyl-DL-Leucine (5 g/day). Results of the clinical ratings are presented for patient 1 (**a**) and patient 2 (**b**). Note: during the study, the motor function assessment in patient 1 was switched from UPDRS III to MDS-UPDRS III. Motor scores are presented without the item action tremor, which does not belong to the cardinal motor signs of Parkinson's disease. Time points of the imaging procedures are indicated as icons. Results of the imaging procedures are presented in Fig. 3 and Table 1. Source data are provided as a Source Data file.

## Dopamine-transporter (DAT) ligand-binding imaging (DAT-SPECT) for evaluation of nigrostriatal function

The DAT-SPECT images and the respective specific striatal binding ratios of patient 1 are presented in Fig. 3a. For example, in her right putamen, the pretreatment DAT-SPECT binding ratio was 1.88 (normal) in 02/2013, decreased to 1.63 (borderline normal) in 03/2014, and then decreased further to 1.22 (pathological) in 01/2019. These data indicate that there was a progressive neurodegeneration of the nigrostriatal dopaminergic projection over 5 years—a hallmark of late prodromal and early PD. In 02/2022, three months after initiation of ADLL treatment, the patient's DAT binding ratio in the right putamen was slightly increased to 1.43. After 22 months of continuous ADLL therapy, the right putaminal DAT binding ratio further increased to 1.67. Thus, oral ADLL therapy partially reversed the decline of the DAT-SPECT binding ratio, almost normalizing it after about 2 years of treatment. Specific DAT-SPECT binding ratios of the right and left striatum, the right and left caudate nucleus, and the left and right putamen are presented for each time point in the lower panel of Fig. 3a. The numeric values of the striatal binding ratios of patient 1 are listed in Table 1.

Patient 2's DAT-SPECT showed a similar pattern of change with ADLL treatment to that observed in patient 1 (Fig. 3b and Table 1). The patient's pretreatment right putaminal DAT-SPECT binding ratio was 1.42 (pathological) in 08/2020. After 18 months of continuous ADLL

treatment, the binding ratio increased to 1.72 in 07/2023—nearly back to normal values (Fig. 3b). The numeric values of the striatal binding ratios of patient 2 are listed in Table 1.

## ¹⁸F-Fluorodeoxyglucose PET imaging (FDG-PET) for evaluation of regional glucose use

For patient 1, the PDRP z-score measured by FDG-PET was 1.72 in 2014, rising to 3.28 in 2018, indicative of a progressive impairment of brain metabolism. In 12/2022, 1 year after starting ADLL therapy, the FDG-PET PDRP-z-score of patient 1 was 3.18—similar to the z-score in 2018 (Fig. 3c, blue triangle). For comparison, 12 other iRBD patients who did not receive ADLL treatment and who had similarly timed FDG-PET scans, were analyzed. Their PDRP-z-scores continued to increase over the 8-year period, indicating disease progression. The average PDRP-z-scores of these 12 untreated iRBD patients that either had phenoconverted (5 subjects; 4 to PD; 1 to DLB) or had not converted (n = 7) in the 8-year follow-up are also shown in Fig. 3c (gray circles; dark gray and light gray shaded areas illustrate the standard deviation). The average z-scores (mean ± standard deviation) of the converter group (n = 5; at baseline in 2014: 2.99 ± 1.68; first follow-up in 2018: 5.75 ± 2.28; second follow-up in 2022: 7.85 ± 2.12) were higher than the average z-scores of the non-converter group (n = 7; at baseline in 2014: 1.63 ± 0.69; first follow-up in 2018: 3.02 ± 1.32; second follow-up in 2022: 4.30 ± 2.04). Thus, when comparing the time course of the PDRP-

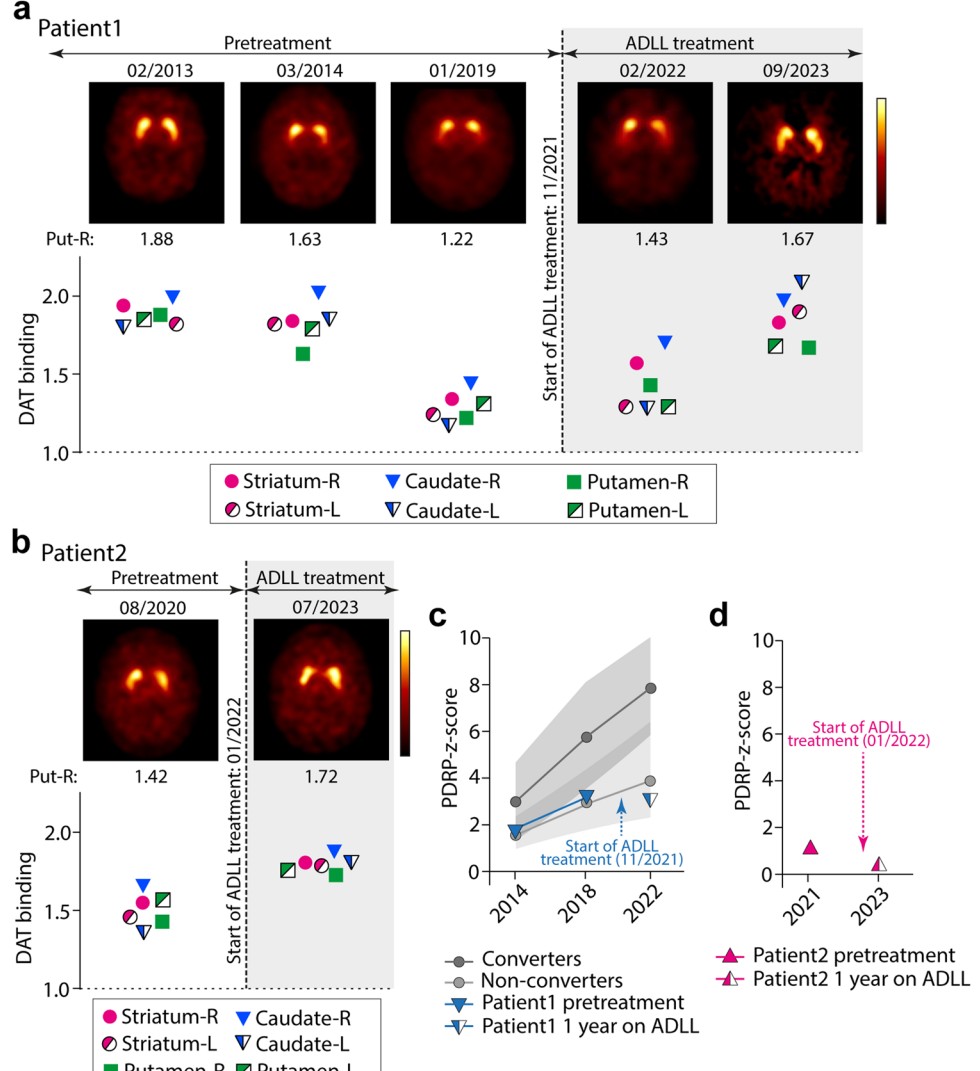

**Fig. 3 | Serial Dopamine -Transporter Ligand-Binding (DAT) SPECT and FDG-PET imaging in both patients. a**, **b** DAT-SPECT results in patients 1 (**a**) and 2 (**b**). The upper part shows representative transversal images of DAT-SPECT at the level of the striatum. For patient 1, three pretreatment images (02/2013, 03/2014, 01/2019) and two images under ADLL therapy (02/2022, 09/2023) are shown. For patient 2, one pretreatment (08/2020) and one image under ADLL therapy (07/2023) are presented. The respective specific-to-non-specific striatal binding ratios of the right putamen are shown directly below the images. The graph shows the specific striatal DAT binding ratios in the striatum, the caudate nucleus and the putamen for the right and left sides. **c** The time course of the values of the z-score of the Parkinson Disease-related-Pattern (PDRP-z-score) in FDG-PET of patient 1 (blue triangle) is shown from 2014 to 2018 and from 2018 to 2022. ADLL treatment was started in November 2021. For patient 1, the PDRP z-score was 1.72 in 2014, rising to 3.28 in 2018. In 12/2022, 1 year after starting ADLL therapy, the FDG-PET PDRP-z-

score was 3.18−similar to the z-score in 2018 (blue triangle). For comparison, the time course of the average PDRP-z-score (mean ± SD, gray shaded area) of 12 untreated iRBD patients from 2014 to 2018 and from 2018 to 2022 are illustrated. This untreated "disease control" iRBD group is divided into the group of converters (n = 5−dark gray circles) and non-converters (n = 7−light gray circles). The average z-scores of the converter group (n = 5; at baseline in 2014: 2.99 ± 1.68; first follow-up in 2018: 5.75 ± 2.28; second follow-up in 2022: 7.85 ± 2.12) were higher than the average z-scores of the non-converter group (n = 7; at baseline in 2014: 1.63 ± 0.69; first follow-up in 2018: 3.02 ± 1.32; second follow-up in 2022: 4.30 ± 2.04). **d** The time course of the PDRP-z-score in FDG-PET of patient 2 (pink triangle) is shown for 06/2021 (1.02). ADLL treatment was started in January 2022. When the second FDG-PET was performed in 01/2023, after 1 year of continuous ADLL treatment, the z-score was reduced to 0.30. Source data are provided as a Source Data file.

z-score of patient 1 (see above) with the PDRP-z-score of the group of untreated iRBD patients between 2018 and 2022, 1 year of ADLL treatment (initiated November 2021) appeared to have stabilized or improved the metabolic activity in PD-related brain areas in patient 1.

In patient 2, his pretreatment PDRP z-score was 1.02 in June 2021 (Fig. 3d). When the second FDG-PET was performed in 01/2023, after 1 year of continuous ADLL treatment, the z-score was reduced to 0.30 (Fig. 3d, pink triangle; see "Methods" for further details). In summary, FDG-PET measures of metabolic activity in the "Parkinson-Disease-related Pattern (PDRP)" stabilized during ADLL treatment.

## Discussion

Here we describe the effects of long-term therapy with ADLL in two subjects suffering from iRBD, hyposmia and cardiac sympathetic denervation−a prodromal phenotype of PD and DLB[3,4]. The beneficial effects of an oral administration of ADLL were reflected in three−therapy-responsive−outcome measures: (1) a marked decline in the subjectively assessed clinical severity of the RBD phenotype (disappearance of aggressive dream content, reduction of dream enactment); (2) a stabilization or reversal of the slow, progressive decline of specific striatal DAT-SPECT binding ratios; and (3) a stabilization or reversal of the expression of the PDRP, an indicator of the progressive

**Table 1 | Striatal DAT-SPECT binding ratios of patient 1 and patient 2**

| Patient 1 | | | | | 11/2021<br>Start ADLL treatment | | Patient 2 | 01/2022<br>Start ADLL treatment |
|---|---|---|---|---|---|---|---|---|
| | 02/2013 | 03/2014 | 04/2016 | 01/2019 | 02/2022<br>3 months under ADLL treatment | 09/2023<br>22 months under ADLL treatment | 08/2020 | 07/2023<br>18 months under ADLL treatment |
| Striatum R | 1.94 | 1.84 | 1.45 | 1.34 | 1.57 | 1.83 | 1.54 | 1.80 |
| Striatum L | 1.82 | 1.82 | 1.39 | 1.24 | 1.29 | 1.90 | 1.45 | 1.78 |
| Caudate R | 1.99 | 2.02 | 1.61 | 1.44 | 1.70 | 1.97 | 1.65 | 1.87 |
| Caudate L | 1.80 | 1.85 | 1.36 | 1.17 | 1.28 | 2.09 | 1.35 | 1.80 |
| Putamen R | 1.88 | 1.63 | 1.27 | 1.22 | 1.43 | 1.67 | 1.42 | 1.72 |
| Putamen L | 1.85 | 1.79 | 1.42 | 1.31 | 1.29 | 1.68 | 1.56 | 1.75 |
| DAT-SPECT scan | Rated normal | Rated normal | Rated ab-normal | Rated ab-normal | Rated ab-normal | Rated normal | Rated ab-normal | Rated borderline normal |
| Performed in | Marburg | Marburg | Marburg | Marburg | Marburg | Amsterdam | Marburg | Marburg |

See also Fig. 3a: DAT-SPECT image of 04/2016 and respective striatal DAT binding ratios are not part of Fig. 3a. All DAT-SPECT scans were analyzed in Amsterdam under blinded conditions. Source data are provided as a Source Data file.

metabolic impairment in the CNS of iRBD and PD—as measured by FDG-PET. On the other hand, long-term therapy with ADLL failed to change the anosmic status of both patients.

The decrease of severity of the RBD-phenotype already after 3 weeks of ADLL therapy may be due to a functional improvement of neurons in circuits controlling REM sleep atonia and dream content[15].

The objective DAT-SPECT and FDG-PET data suggest that ADLL treatment may modify the progression of iRBD and its conversion to the PD phenotype. Striatal DAT-SPECT measures the abundance of the plasma membrane dopamine transporter (DAT) located in the axon terminals of nigrostriatal dopaminergic neurons. Loss of these axons has long been linked to the emergence of PD symptoms[16]. In animal models of PD, there is downregulation of axonal DAT expression prior to actual neurodegeneration[17]. The stabilization or improvement of DAT-SPECT binding ratios in ADLL-treated patients suggests that treatment might have "rescued" nigrostriatal neuronal somata and/or axons. In respect to the second imaging procedure FDG-PET, the PDRP-z-score—an overall indicator of metabolic CNS changes—steadily increases in iRBD patients prior to conversion from iRBD to manifest PD[18,19]. Thus, the observed stabilization or improvement of the PDRP-z-scores based on FDG-PET measurements suggest that ADLL treatment halted the progressive network dysfunction which in part reflects the loss of dopaminergic axons[20]. This conclusion was supported by the longitudinal comparison of the FDG-PET PDRP-z-scores of the ADLL-treated patient 1 with those of the 12 untreated iRBD patients. It remains to be shown in controlled long-term follow-up studies whether these observed changes in the FDG-PET-derived PDRP will be stable over time. On the other hand, our data support previous statements that both imaging procedures may be useful prodromal progression markers for PD[21] and—according to the presented data—appear to be therapy-responsive.

Acetyl-leucine is taken up by monocarboxylate transporters with a high transport capacity, that are ubiquitously expressed[22]. It has several mechanisms of action. Animal studies showed that AL improves lysosomal function, metabolic flux and adenosine triphosphate production[7] as well as neuronal activity[23,24] (for further details, see Section A in Supplementary Information). Recently, in an experimental mouse model of prodromal PD, we studied the effect of preformed fibrils of alpha-synuclein—locally applied—on neurons in the PD-vulnerable structures substantia nigra (SN) and pedunculopontine nucleus (PPN). In both the dopaminergic SN and cholinergic PPN neurons, the exposure to aggregated alpha-synuclein led to a marked decrease of intraneuronal ATP production and impaired lysosomal function[14]. In addition, our recently published phase 3 trial in patients

suffering from Niemann-Pick disease type C confirmed the beneficial effect of acetyl-leucine in this devastating neurodegenerative lysosomal storage disorder[8]. Thus, AL might provide a multimodal improvement in neuronal function and therefore neuroprotection. These effects may extend to prodromal PD.

Our report has obvious limitations. Due to regulations on "individual cases of off-label use" we investigated only two iRBD patients with open label ADLL therapy without a placebo control. We used the available racemate of ADLL instead of the bioactive enantiomer acetyl-L-leucine, which is not currently available[25]. The clinical outcome measure is patient-centered and based on the subjective daily rating of the RBD severity by patient and spouse. In addition, a protocol with defined times for imaging assessment was not employed. Whereas DAT-SPECT is a well-studied and widely accepted imaging method to monitor the PD-prodromal nigrostriatal neurodegeneration, the PDRP in the FDG-PET method is still discussed as a prodromal progression marker and thus should be considered as a supportive parameter to the results obtained with DAT-SPECT. For further discussion of limitations, see Section E in Supplementary Information.

In conclusion, the reduction of the severity of the RBD phenotype may reflect a symptomatic effect. The observed changes in the two objective imaging progression markers, however, raise the possibility that ADLL treatment slowed the progression to the PD phenotype, i.e., it might reflect a disease-modifying effect of ADLL in prodromal PD. Thus, these results support further explorations whether ADLL may have disease-modifying properties in prodromal PD. They also provide a compelling rationale for a placebo-controlled trial in iRBD patients with bioactive acetyl-L-leucine.

## Methods

### Participants in the individual case of off-label use study

Two patients with video-polysomnography (PSG) confirmed diagnosis of iRBD[26] gave written informed consent to receive therapy with ADLL under "individual case of off-label use rules"— according to CARE guidelines and in compliance with the Declaration of Helsinki principles.

Our research complies with all relevant ethical regulations. Ethical approval for individual case of off-label use was given by the Ethics Committee, Faculty of Medicine, Philipps-University Marburg (Chairperson of the committee: Prof. Carola Seifart, MD). As this article is a case report, no statistical method was used to predetermine sample size. No data were excluded from the analyses. The experiments were not randomized. Apart from the unblinded treating physician (first author), all investigators were blinded to allocation during experiments and outcome assessment.

Patients were treated with oral ADLL (5 g per day)[6,7,9,10] for a period of >18 months. Both patients were otherwise healthy. They were also participants in a parallel long-term observational natural history study called REMPET: Rapid-Eye-Movement (REM) Sleep Behavior Disorder and Fluorodeoxyglucose Positron Emission Tomography (FDG-PET). The study REMPET investigates the progression of the metabolic "Parkinson-Disease-related-Pattern (PDRP)" in FDG-PET of iRBD-patients over 10 years[18]. As part of the REMPET study, both patients received serial molecular imaging with DAT-SPECT and FDG-PET, annual clinical monitoring, and a cardiac [123]I-metaiodobenzylguanidine scintigraphy ([123]I-MIBG) scan[4,19,27] (patient 1 received the MIBG scan at baseline in 2014, 9 years before treatment with ADLL; patient 2 received the MIBG scan in 2022, under ADLL treatment) showing in both patients a reduction in cardiac sympathetic innervation. In addition, their olfactory function was tested with the Sniffin Sticks method (Threshold (T), Discrimination (D), Identification (I)) which provides a TDI-sum-score on olfactory function (see below)[4,19,28,29].

## Clinical evaluation

Both patients receiving ADLL therapy were evaluated with the Unified Parkinson Disease Rating Scale (UPDRS−original version; motor part: UPDRS III−range from 0 to 104, higher scores indicate greater impairment) or the Movement Disorder Society (MDS)-UPDRS (revised version; motor part: MDS-UPDRS III−range from 0 to 137, higher scores indicate greater impairment)[30,31]. Both scores include the item "action tremor", which is not a cardinal motor sign of PD. We therefore calculated the UPDRS III and the MDS-UDPDRS III with and without the item "action tremor". The clinical evaluation also included the screening test Montreal Cognitive Assessment (MoCA−range from 0 to 30, higher score indicating better cognitive performance; the threshold for mild cognitive impairment was defined below 26)[32] and the SCOPA-AUT (Non-motor Autonomic Symptoms Questionnaire−range from 0 to 69, higher scores signify greater impairment of autonomic functions)[33] at baseline and in about 6–12 month intervals for follow-up−all performed by blinded personnel.

**Patient 1.** In 2013, a female subject reported symptoms of RBD starting in 2006. PSG[26] confirmed the diagnosis of iRBD in 2011. For the symptomatic treatment of iRBD, she had been prescribed a daily dose of 1 mg clonazepam at night since 2018. She underwent four pre-treatment DAT-SPECT scans ([123]I-FP-CIT SPECT, marketed as DaTSCAN) in 02/2013, 03/2014, 04/2016, and 01/2019. In the REMPET study, two pretreatment FDG-PETs were performed in 2014 and 2018. In November 2021, she began taking 5 g/d ADLL, i.e., about three years after the last "baseline" DAT-SPECT in 2019 and after the second FDG-PET in 2018. After 3 months of ADLL treatment, she received the next DAT-SPECT scan in February 2022 and a further DAT-SPECT scan after 22 months of continuous ADLL therapy in September 2023. Likewise, after taking 13 months of continuous ADLL therapy, she underwent a third FDG-PET in December 2022.

**Patient 2.** A male subject reported RBD symptoms in 2018. PSG[26] confirmed the diagnosis in June 2020. He was not taking any symptomatic therapy for iRBD nor any other concomitant medication. He received a baseline DAT-SPECT scan in August 2020 and an FDG-PET scan in June 2021, 17 and 7 months before starting the continuous ADLL therapy (5 g/d) in January 2022. After 12 months of continuous ADLL therapy, he underwent a second FDG-PET scan in January 2023 and after 18 months of continuous ADLL therapy a second DAT-SPECT scan in July 2023.

## Treatment regime with ADLL

ADLL is commercially available under the trade name Tanganil® in France. It has been registered for the indication "vertigo" since 1960. The drug contains the racemate of acetyl-leucine, i.e., the inactive D-form and the bioactive enantiomer, the L-form of acetyl-leucine in equal parts. The daily dosage of ADLL in previous clinical studies for other indications was 5 g/day (2 g in the morning, 1.5 g at noon and 1.5 g at night)[6,9,10]. This dosage is similar to the equivalent dosage in animal studies (0.1 g per kg and day)[7]. For this case report, we administered the already investigated total oral daily dosage of 5 g/day. However, the distribution throughout the day was changed and the highest dose was taken in the evening with a recommended dosage of 1 g in the morning, 1–1.5 g at noon and 2.5–3 g in the late evening. If considered necessary, the dosage was slightly changed during the treatment period of 18–22 months. The highest ADLL dose taken was 5.5 g with 1 g in the morning and at noon and 3.5 gram in the late evening (patient 1, see Section C in Supplementary Information).

## "Disease control" untreated iRBD patients− [18]F-Fluorodeoxyglucose-PET

For comparison, the FDG-PET scans generated in 2014, 2018, and 2022 from 12 other iRBD participants in the REMPET study, who did not receive ADLL, were utilized and analyzed in parallel to the three FDG-PET scans of patient 1 (Fig. 3c, gray circles; see section FDG-PET below). All these "control" untreated iRBD patients had a marked hyposmia or anosmia at baseline in 2014 and in 9 of these 12 iRBD patients an MIBG scan was performed in 2014 and showed a reduced cardiac sympathetic innervation, an indicator that the respective iRBD patients presented a prodromal stage of PD[4].

Phenoconversion of these untreated iRBD patients to PD, DLB or (rarely) multiple system atrophy (MSA) was searched for and either excluded or confirmed by a neurologist (AJ) according to the published diagnostic criteria[34,35].

## Inclusion criteria for all participating iRBD patients

The diagnosis made by video-assisted polysomnography (vPSG) was mandatory. Patients were informed that they would be allowed to take symptomatic therapy for RBD−such as clonazepam or melatonin in normal or slow-release preparations−if requested or necessary during the study at the discretion of the principal investigator.

## Exclusion criteria for all participating iRBD patients

The presence of symptoms and signs for a manifest PD, DLB or MSA or any other neurological or psychiatric disorder were exclusionary. Cognitive impairment, as defined by a MoCA score below 26 at baseline, was exclusionary. The iRBD patients did not suffer from the following diseases: heart/kidney failure, myocardial infarction in the last five years, diabetes, amyloid or other neuropathy, pheochromocytoma. They were not taking any medications (reserpine, opioids, labetalol, phenylpropanolamine, phenylephrine), which might affect [123]I-MIBG results.

## Sniffin' Sticks olfactory function test

Olfactory function was assessed at baseline and at most dates of the imaging procedures (Fig. 2) with the full range Sniffin' Sticks method consisting of a test for threshold of odor recognition (T), a test for discrimination between odors (D), and a test for identification of odors (I)[4,19,28]. Each subtest results in a subscore T, D, and I, respectively, and the sum of all three subscores is presented as the TDI sum-score (range 0–48, TDI sum-score of ≥32 was defined as normal, TDI sum-score between 16 and 31 was defined as hyposmia and anosmia was defined as a TDI sum-score ≤15 (see Section D in Supplementary Information for test details and Fig. 2a, b as well as Supplementary Table 1 for results). During the olfactory testing procedure, the patients were blindfolded. The olfactory function test was performed by a specially trained study nurse who was blinded for the purpose of the investigation[4,28,29]. The TDI sum-score is not agreed as a prodromal progression marker in RBD[4] and has not been shown so far to be therapy-responsive. Therefore, the TDI sum-score is considered as an exploratory measure.

## Clinical outcome Measure

As a clinical outcome parameter for measuring the severity of the subjective RBD phenotype, we used an RBD severity sum-score. After the start of ADLL treatment, the severity and frequency of RBD symptoms (dream enactment) were recorded daily in the morning by the patient and spouse in a standardized form according to a modified version of a published RBD diary[36]. The modified RBD diary consists of two separate parts: (1) severity - daily ranking scale ranging from 0 to 4 (0−no RBD symptoms; 1−speaking or mild jerks; 2−shouting or complex, non-aggressive movements; 3−aggressive movements with the risk of injuring her-/himself or the partner; 4−movements, which are so violent that the person falls out of bed); (2) frequency - weekly ranking scale ranging from 0 to 3 (0−no RBD symptoms; 1−RBD symptoms 1 to 2 nights per week; 2−RBD symptoms 3 to 5 nights per week; 3−RBD symptoms 6 to 7 days per week). Filled-out forms covering 7 daily ratings of the week were sent to the principal investigator every Sunday evening.

For this study, daily scores were analyzed for a period of 561 days. The daily scores of the first 18 weeks and the last 18 weeks are presented in Supplementary Fig. 1 (see Section C in Supplementary Information). To present the data in Fig. 1 of the main manuscript, we added up the daily RBD severity scores of 21 consecutive days to create a 3-week RBD-severity sum-score (RBD-SS-3−range from 0 to 84; the higher the score, the greater the severity).

## Imaging outcome measures

### Dopamine-transporter (DAT) ligand-binding imaging (DAT-SPECT).

As key outcome parameter we employed serial presynaptic dopamine-transporter (DAT) ligand-binding SPECT-imaging by $^{123}$I-2β-carbo-methoxy-3β-(4-iodophenyl)-N-(3-fluoropropyl)-nortropane ($^{123}$I-FP-CIT) (DAT-SPECT) to monitor the dopaminergic nigrostriatal synaptic density in the two patients who received ADLL therapy.

Patient 1 underwent pretreatment DAT-SPECT scans in 02/2013, 03/2014, 04/2016, and 01/2019. The DAT-SPECT scan in 01/2019 was selected as the pretreatment "baseline image" for further comparisons. The absolute time interval between the acquisition of the "baseline" DAT-SPECT scan in January 2019 and initiation of ADLL therapy in November 2021 was 34 months. Patient 1 received the first DAT-SPECT scan "under ADLL therapy" after 3 months of ADLL therapy in February 2022. The second DAT-SPECT scan under continuous ADLL therapy was performed in September 2023, that is 22 months after the initiation of ADLL therapy. In patient 1, ADLL was withdrawn for 10 days before the DAT-SPECT "under ADLL therapy" (plasma terminal half-life ($t_{1/2}$) of the active enantiomer Acetyl-L-Leucine is 0.96 (±0.18) hours) in order to avoid any interference of the drug per se with the DAT-SPECT procedure.

Patient 2 received a "baseline" DAT-SPECT scan in August 2020. The absolute time interval between acquisition of the pretreatment "baseline" DAT-SPECT scan in 2020 and initiation of ADLL therapy in January 2022 was 17 months. He received his DAT-SPECT scan "under ADLL therapy" in July 2023, i.e., 18 months after the start of the continuous ADLL therapy. Due to miscommunication, patient 2 did not withdraw ADLL for the DAT-SPECT scan in July 2023.

The DAT-SPECT procedure including the acquisition and reconstructions of the DAT-SPECT scans has been published[37–39] (see also Section D in Supplementary Information). Apart from the last DAT-SPECT of patient 1 performed in 09/2023, all DAT-SPECT investigations were performed in the Department of Nuclear Medicine, University of Marburg. Thus, the DAT-SPECT scans of patient 1 in 2013, 2014, 2016, 2019, and 2022 and the DAT SPECT scans of patient 2 in 2020 and 2023 were carried out on the same SPECT camera. After receiving the results of the second DAT-SPECT scan of patient 2 (showing an increase in the specific striatal DAT binding ratios in 07/2023 (1.72, right putamen) in comparison to the DAT-SPECT in 08/2020 (1.42, right putamen) (Fig. 3b and Table 1), we decided to perform the next and last DAT-

SPECT scan in patient 1 in a different Department of Nuclear Medicine under blinded conditions. Patient 1 therefore agreed to travel to the Department of Radiology and Nuclear Medicine, University of Amsterdam, Amsterdam UMC (Prof. J. Booij). No one on the team at the Department of Nuclear Medicine in Amsterdam had any information about the purpose of the DAT-SPECT appointment. In fact, the patient was thought to be undergoing a routine DAT-SPECT investigation as part of her diagnostic work-up[37].

After the last DAT-SPECT scan of patient 1 had been performed in Amsterdam, the group of Prof. J. Booij received all reconstructed data files of all previous DAT-SPECT investigations−generated in Marburg – without any further information about the purpose of the study.

All DAT-SPECT scans were subsequently analyzed by blinded personnel at the Department of Radiology and Nuclear Medicine, University of Amsterdam, Amsterdam UMC (Prof. J. Booij), The Netherlands, using the Brain Registration & Analysis Software Suite (BRASS; HERMES Medical, Sweden). Specific striatal binding ratios were determined and corrected for age as described earlier[38,39]. Specific to non-specific binding ratios were calculated in the striatum, the caudate nucleus and the putamen for each site, using the occipital cortex as a reference to assess non-specific binding. The numeric values of the striatal binding ratios are listed in Table 1. Transverse images for Fig. 3 were reconstructed as previously reported[18].

### $^{18}$F-Fluorodeoxyglucose PET imaging (FDG-PET).

We employed serial $^{18}$F-Fluorodeoxyglucose PET imaging (FDG-PET) to identify the pathologic metabolic "Parkinson-Disease-related -Pattern (PDRP)" over time. Expression-z-scores for the PDRP were quantified in all scans (higher z-score value indicating stronger expression of PDRP). In order to minimize variation in the FDG-PET procedure, patient 1 (but not patient 2 (see below)), and all other "disease control" untreated iRBD patients underwent FDG-PET imaging at the Department of Nuclear Medicine and Molecular Imaging, University Medical Center Groningen (UMCG), The Netherlands.

### FDG-PET analysis of 12 untreated iRBD patients for reference.

FDG-PET scanning was performed for 12 untreated iRBD patients at baseline, first follow-up (approximately 4 years after baseline) and second follow-up (approximately 8 years after baseline). During the 8-year course of the study, 4 iRBD patients converted to manifest Parkinson's disease (PD), 1 to manifest Dementia with Lewy bodies (DLB) and 7 remained in the prodromal stage iRBD and were thus defined as "non-converters".

All baseline and follow-up scans were performed on a Siemens Biograph mCT64 or mCT40 PET/CT camera (Siemens, Munich, Germany) at the Department of Nuclear Medicine and Molecular Imaging, University Medical Center of Groningen (UMCG), The Netherlands, using a static imaging protocol. Images were reconstructed with OSEM3D (3 iterations, 21 subsets), time-of-flight, point-spread-function, and smoothed with a Gaussian 8-mm full-width-at-half-maximum spatial filter according to the EANM guidelines[18,40]. The matrix size was 256 (corresponding to a voxel size of 2 mm × 3.18 mm × 3.18 mm).

Central nervous system depressants and any RBD-related medications (i.e., melatonin or clonazepam) were discontinued in all subjects for at least 24 hours before scanning. Likewise, in the 5 iRBD patients, who had converted to manifest PD or DLB during the 8-year period of the REMPET study, levodopa or dopamine agonists were discontinued at least 24 h before scanning.

All scans were spatially normalized to an FDG-PET template in Montreal Neurological Institute brain space[18,41] using SPM12 software (Welcome Centre for Human Neuroimaging, London, UK) implemented in MATLAB (version R2019a; MathWorks, Natick, MA, USA)[40].

The Parkinson's disease-related pattern (PDRP) that was used in this study was previously identified in FDG-PET scans of a cohort of 19 PD patients (13 M/6 F, age 63.9 ± 7.8), in the off-levodopa state, in

comparison to a cohort of 17 healthy controls (12 M/5 F, age 61.5 ± 7.5 years)[41]. PDRP subject scores were calculated for each scan[18]. For this study, PDRP subject scores were z-scored to a cohort of 12 age- and gender-matched healthy controls (10 male/2 female, age 65.96 ± 6.21 years). These controls only underwent baseline FDG-PET imaging. By definition, healthy controls had an average z-score of 0, with a standard deviation of 1.

Average PDRP-z-scores (mean ± SD) of the iRBD patients for baseline, follow-up 1 and follow-up 2 visits were plotted for converters ($n = 5$) and non-converters ($n = 7$) separately (Fig. 3c).

**FDG-PET in patient 1**. Patient 1 was part of the REMPET3 study. FDG-PET scans were performed as described above in Groningen, with an identical protocol.

**FDG-PET in patient 2**. Patient 2 also participated in the REMPET3 study. FDG-PET imaging was, however, performed at the Department of Nuclear Medicine, University of Marburg on a Siemens Biograph 6—and not at the UMCG in Groningen—using a static imaging protocol. Images were reconstructed with OSEM3D (3 iterations, 21 subsets), point-spread-function, and smoothed with a Gaussian 4-mm full-width-at-half-maximum spatial filter. The matrix size was 336 and Zoom in position 2. Reconstructed FDG-PET data were sent to the UMCG, Groningen, The Netherlands for further analysis (see above)[37,40].

It is well known that variations in scanners and image reconstruction algorithms can impact PDRP expression scores. One way to resolve this is to apply a z-transformation to healthy control data from the same camera with an identical reconstruction protocol[18,37].

However, FDG-PET scans of healthy controls were not available from the Marburg site. Therefore, the FDG-PET scans of patient 2 were preprocessed in a similar way as the UMCG data, and the PDRP was calculated in the same manner, using the same UMCG healthy control cohort ($n = 12$) for z-scoring. As a consequence, the z-scores for patient 2 cannot be compared to the scores of the reference iRBD cohort ($n = 12$) or of patient 1. Nevertheless, because any noise from a difference in acquisition protocols is systematically present in both scans, the difference in PDRP-z-scores between the two scans of patient 2 can still be appreciated. After initiation of ADLL therapy in 01/2022, the PDRP-z-score decreased from 1.02 (06/2021) to 0.30 (01/2023).

In both patients, ADLL was withdrawn for 10 days before the FDG-PET "under ADLL therapy" in order to avoid any interference of the drug per se with the FDG-PET procedure.

**[123I]-Metaiodobenzylguanidine cardiac scintigraphy (MIBG)**
Patient 1 receiving ADLL therapy and 9 out of 12 disease control iRBD participants underwent cardiac [123I]MIBG scintigraphy at baseline performed according to the standard operating procedures (see Section D in Supplementary Information) of the Department of Nuclear Medicine, Marburg, Germany. Patient 2 received the MIBG investigation under ADLL therapy in May 2022. Regions of interest (ROI) were manually placed on planar anterior images. A rectangular ROI was used for the mediastinum and a circular ROI for the left ventricle of the heart. According to the in-house code, a heart-to-mediastinum ratio of [123I]MIBG-binding of <1.5 was considered pathological[4].

**Statistical reporting section**
This article reports two cases. Therefore, data of the RBD severity scores RBD-SS and striatal binding ratios for the serial DAT-SPECT images are presented as individual values. No further statistical analysis was performed. z-scores of the Parkinson-Disease-related-Pattern (PDRP) of the FDG-PET images were calculated according to published algorithms[18,37]. Mean ± standard deviation were calculated for the z-scores of the converters and the non-converters, respectively, in the untreated "disease control" group of iRBD patients. Data analysis was

performed using GraphPad Prism (version 8.3.1 GraphPad Software, USA). All figures were created with Adobe Illustrator version 25.1 (Adobe Systems).

**Reporting summary**
Further information on research design is available in the Nature Portfolio Reporting Summary linked to this article.

## Data availability
All raw individual research data are available in the Main Manuscript and the Source Data File. All measures have been taken to protect the patients' identity. Data can be shared publicly as participants consented to the sharing of their data as per European Union's General Data Protection Regulation (EU GDPR) and the corresponding German privacy laws. The original study design and data including de-identified participant data sets are made available to researchers without any restriction. Raw source data are provided with this article in a separate Source Data File. Source data are provided with this paper.

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

## Acknowledgements

Funding for this study was in part provided by the Andre-Leysen-Ulran-Foundation, Belgium (W.H.O.), the ParkinsonFonds, Deutschland (W.H.O. and A.J.) and Stichting ParkinsonFonds, The Netherlands (K.L. and S.K.M.) and the JPB Foundation (D.J.S.). W.H.O. is Hertie Senior Research Professor supported by the Charitable Hertie-Foundation, Frankfurt/Main, Germany. The authors are very grateful to the two individuals suffering from iRBD who agreed to participate in this study under "individual case of off-label use rules". The expert support in olfactory function testing by Christine Höft, Marburg, and the expert support in analyzing the FDG-PET data by Anna Dortmond, Groningen, are highly appreciated. The authors thank Mrs. Taylor Fields, IntraBio Ltd., London, United Kingdom for a critical review of the manuscript.

## Author contributions

W.H.O. conceptualized, planned and oversaw all aspects of the study, which included obtaining consensus from the subjects to participate in the individual case of off-label use. A.J. analyzed the sleep lab data, diagnosed the iRBD subjects, generated the clinical data and collected the images, and interpreted the data. M.T.H. and F.F.G. conceptualized the study, performed the statistical analysis and designed the figures. E.S. planned and coordinated the study and acquired the clinical and technical (olfactory function test) data and collected the images. S.K.M., G.C., K.L., and J.B. analyzed the DAT-SPECT and FDG-PET imaging data and interpreted the imaging data. D.J.S. conceptualized the study and interpreted the data. L.T. reevaluated the iRBD diagnosis and provided substantial contributions to the clinical data analysis. M.S. conceptualized the study and interpreted the data. W.H.O., A.J., F.F., M.H., and M.S. wrote the supplementary information. W.H.O. and M.S. wrote the main manuscript with input and substantial revisions from all authors.

## Funding

## Competing interests

Wolfgang H. Oertel has received speaker's honoria on educational symposia sponsored by Abbvie, the International Movement Disorders Society and Stada Pharma. He acts as a consultant for Lario Therapeutics and is a member of advisory boards with Intrabio and MODAG. He holds stock options with Intrabio related to this manuscript and stock options with MODAG unrelated to this work. The institution of W.H.O., not W.H.O personally received/s scientific grants from the German Research Foundation, the Michael J Fox Foundation and Rittal Foundation unrelated to the manuscript. Jan Booij is a consultant of GE Healthcare. The institution of J.B., not J.B. personally received research funding from GE Healthcare. Lars Timmermann has received speaker's honoria on educational symposia sponsored by Abbvie, Boston Scientific, DIAPLAN, Neuraxpharm, Novartis, the International Movement Disorders Society

und Teva. He has been a consultant for Boston Scientific. The institution of L.T., not L.T. personally received/s funding by Boston Scientific, the German Research Foundation, the German Ministry of Education and Research, the Otto-Loewi-Foundation and the Deutsche Parkinson Vereinigung. Neither L.T. nor any member of his family holds stocks, stock options, patents or financial interests in any of the above-mentioned companies or their competitors. Michael Strupp is Joint Chief Editor of the Journal of Neurology, Editor in Chief of Frontiers of Neuro-otology and Section Editor of F1000. He has received speaker's honoraria on educational symposia sponsored by Abbott, Auris Medical, Biogen, Eisai, Grünenthal, GSK, Henning Pharma, Interacoustics, J&J, MSD, NeuroUpdate, Otometrics, Pierre-Fabre, TEVA, UCB, and Viatris. He acts as a consultant for Abbott, AurisMedical, Bulbitec, Heel, IntraBio, Sensorion and Vertify. He is an investor and share-holder of IntraBio. The institution of M.S., not M.S. personally received/s support for clinical studies from Decibel, USA, Cure within Reach, USA and Heel, Germany. He distributes "M-glasses" and "Positional vertigo App". The remaining authors declare no competing interests.

## Additional information

[1]Department of Neurology, Philipps University of Marburg, Marburg, Germany. [2]Institute of Neurogenomics, Helmholtz Center for Medicine and Environment, Munich, Germany. [3]Department of Psychiatry and Psychotherapy, Philipps University of Marburg, Marburg, Germany. [4]Department of Neuroscience, Feinberg School of Medicine, Northwestern University, Chicago, IL, USA. [5]Department of Neurology, University Medical Center Groningen, Groningen, The Netherlands. [6]Department of Nuclear Medicine and Molecular Imaging, University Medical Center Groningen, Groningen, The Netherlands. [7]Department of Radiology and Nuclear Medicine, Amsterdam University Medical Centers, University of Amsterdam, Amsterdam, The Netherlands. [8]Department of Neurology, LMU University Hospital, LMU, Munich, Germany. ✉e-mail: oertelw@med.uni-marburg.de; Michael.Strupp@med.uni-muenchen.de

