## [Peer Review File · Nature Communications]

Acetyl-DL-leucine in two individuals with REM sleep behavior disorder improves symptoms, reverses loss of striatal dopamine-transporter binding and stabilizes pathological metabolic brain pattern - case reportsREVIEWER COMMENTS

Reviewer #1 (Remarks to the Author):

Thanks for asking me to review this manuscript. The authors describe 2 case reports of individuals with RBD who were treated with Acetyl D,L Leucine. One had moderate RBD symptoms, the other mild. They describe a marked and sustained improvement in the RBD scores albeit the patient with mild RBD later had an acute worsening associated with alcohol use.

Neither patient had motor or cognitive features of PD, but both had abnormal MIBG scores.

Most intriguing is the improvement in DATSPECT scores seen in both patients temporally associated with ADLL use. The authors also collected FDG PET data. To try and disentangle whether there were any effects of ADLL, they retrospectively compared the change in these 2 individuals to the change seen in a group of non treated RBD patients. While the mean data are of interest, the number of patients and the lack of randomisation or protocolisation makes needs to be considered when making this comparison.

This report is of interest particularly given the recently published phase 3 trial of this agent in patients with NPC.

I have a number of comments.

The final sentence in the Abstract needs to be moderated- these findings are highly preliminary. It would be better to say “support exploration whether ADLL may have disease modifying properties in prodromal PD”.

The authors need to be more cautious, and offer more critique. The MOCA worsened in both patients and they have remained hyposmic. The MDS UPDRS scores is unchanged. It is not clear why the authors have presented the data without tremor.

I would suggest Table 1 is converted into 2 Figures (patient 1 then 2) which shows a timeline, below which the scores on various metrics can be shown in comparison to the intervention timing. This would enable a reduction in the number of words describing the imaging timing in the Methods. The details regarding the imaging acquisition could be made more succinct. Most important is the point that the team have use the same SPECT camera to collect the images.

Table 2 is unnecessary. Tables 3 and 4 could be combined.

Page 6 line 134- clarify that the date 2022 refers to the cardiac MIBG scan

The drug was withdrawn 10 days before the DATSCAN imaging to prevent any interaction between

drug and DATSCAN binding. While the half-life of ADLL in the blood is short, the authors need to discuss what (little) evidence there is that this is sufficient time for the drug to be cleared from the CNS.

Reviewer #2 (Remarks to the Author):

The authors describe an outstanding effect of N-Acetyl -DL-Leucine on Rem Sleep Behavior Disorders of two patients with Parkinson Disease, which was accompanied by increment of putaminal binding ratio of DAT-SCAN SPECT and stabilization of FDG-PET binding (measured according to the ParkinsonDiseaseRelatedPattern-PDRP evaluation). The work is extremely relevant for PD patients.

The works will need additional evidence, as it could be performed in two patients only, but it could provide a starting point for further research.

Methodologically the work is sound, clinical and Imaging quantifications are state of art, I could find no flaws in data analysis, the work will be easily reproduceable by other experts.

English language is adequate. The authors anticipate the analysis of limitations appropriately.

NCOMMS-24-10685-A

Response to the remarks and critiques of the reviewers

Dear Reviewers

Thank you very much for your precious time and constructive criticisms. A point-by-point response to your critiques is attached.

We hope we have addressed all the major concerns expressed. Our replies are marked in **blue**. The changes and additions in the manuscript are highlighted in **yellow**.

Newly added sentences are highlighted in yellow.

Reviewer #1 (Remarks to the Author):

#Remark 1:

Thanks for asking me to review this manuscript. The authors describe 2 case reports of individuals with RBD who were treated with Acetyl D,L Leucine. One had moderate RBD symptoms, the other mild. They describe a marked and sustained improvement in the RBD scores albeit the patient with mild RBD later had an acute worsening associated with alcohol use. Neither patient had motor or cognitive features of PD, but both had abnormal MIBG scores.

RESPONSE: We confirm that the second individual more mildly affected by isolated RBD (iRBD) showed aggravation of the RBD phenotype by intake of alcohol - even under Acetyl-DL-Leucine therapy. This phenomenon is well known in the literature, especially in male subjects suffering from iRBD (Jun, J.-S. et al. *Nature and science of sleep* 14, 1713–1720; 10.2147/NSS.S372823 (2022)). The female patient does not drink alcohol.

#Remark 2:

Most intriguing is the improvement in DATSPECT scores seen in both patients temporally associated with ADLL use. The authors also collected FDG PET data. To try and disentangle whether there were any effects of ADLL, they retrospectively compared the change in these 2 individuals to the change seen in a group of non-treated RBD patients. While the mean data are of interest, the number of patients and the lack of randomisation or protocolisation makes needs to be considered when making this comparison.

RESPONSE: We agree that the relevant objective finding of our pilot study is the improvement in DAT-SPECT scores over time under continuous ADLL therapy.

We also agree that the additional analysis of the z-score of the Parkinson disease related pattern in Fluorodeoxyglucose PET in the two ADLL-treated iRBD subjects provides a supportive data set and needs to be confirmed in a larger number of iRBD patients in a randomized placebo-controlled strictly protocolized trial.

We have therefore modified the wording in the evaluation of the FDG-PET findings in the Discussion.

The respective paragraph in the Discussion on page 8/9 now reads as follows:

In respect to the second imaging procedure FDG-PET, the PDRP-z-score – an overall indicator of metabolic CNS changes – steadily increases in iRBD patients prior to conversion from iRBD to manifest PD^{16,17}. Thus, the observed stabilization or improvement of the PDRP-z-scores based on FDG-PET measurements suggest that ADLL-treatment halted the progressive network dysfunction which in part reflects the loss of dopaminergic axons³⁶. This conclusion was supported by the longitudinal comparison of the FDG-PET PDRP-z-scores of the ADLL-treated patient 1 with those of the 12 untreated iRBD patients. It remains, however, to be shown in controlled long-term follow-up studies whether these observed changes in the FDG-PET derived PDRP will be stable over time. On the other hand, our data support previous statements that both employed imaging procedures may be useful prodromal progression markers for PD⁴¹ and – according to the presented data – appear to be therapy-responsive.

The respective paragraph in the Discussion on page 9 now reads as follows:

“[...] Whereas DAT-SPECT is a well-studied and widely accepted imaging method to monitor the PD-prodromal nigrostriatal neurodegeneration, the PDRP in the FDG-PET method is still discussed as a prodromal progression marker and thus should be considered as a supportive parameter to the results obtained with DAT-SPECT. For further discussion of limitations, see Supplementary Material.”

#Remark 3:

This report is of interest particularly given the recently published phase 3 trial of this agent in patients with NPC.

RESPONSE: We fully agree.

#Remark 4:

The final sentence in the Abstract needs to be moderated- these findings are highly preliminary. It would be better to say “support exploration whether ADLL may have disease modifying properties in prodromal PD”.

RESPONSE: We are grateful for this recommendation. We agree that we present highly preliminary data. We have therefore changed the final sentence in the Abstract according to the advice of reviewer 1.

The respective sentence in the Abstract on page 3 reads now as follows:

“[...] These results support further explorations whether acetyl-leucine may have disease modifying properties in prodromal PD.”

The respective sentence in the Conclusion of the Discussion on page 10 likewise has been modified as follows:

Thus, these results support further explorations whether ADLL may have disease-modifying properties in prodromal PD. They also provide a compelling rationale for a placebo-controlled trial in iRBD patients with bioactive acetyl-L-leucine.

#Remark 5:

The authors need to be more cautious, and offer more critique. The MOCA worsened in both patients and they have remained hyposmic. The MDS UPDRS scores is unchanged. It is not clear why the authors have presented the data without tremor.

RESPONSE: We thank the reviewer for discussing the findings of the MoCA and olfactory function test as well as the presentation of the UPDRS III (motor) and MDS-UPDRS III (motor) subscores with and without action tremor.

The MoCA test is a screening test and prone to a variability over time. Its results depend on the level of wakefulness and/or attention. This fact is reflected by the graph, for example, of patient 1, who scored 20 (i.e. clearly in the pathological range) in 2013. The following test results were in the normal range before the initiation of therapy with ADLL. We agree that during the period of ADLL therapy there is a tendency for the MoCA score of patient 1 to worsen. We therefore reinvited (February 2024 – still under ADLL therapy) patient 1 for a new MoCA assessment and the score was 24, which is, by definition of the inclusion criteria, in the range of a mild cognitive impairment. Thus, we cannot rule out that this now 80-year-old RBD patient will develop cognitive impairment in the future. In patient 2 (now 59 years of age), there is also a trend towards lower normal MoCA values during the period of the ADLL therapy. A recent MoCA test (March 2024 – still under ADLL therapy) revealed a score of 28 – this finding is in line with the stated variation in the results of the MoCA screening test.

In respect to the olfactory function test, we employed the full range Sniffin Sticks method. This test is very demanding and takes at least one hour to complete. Again, the result depends on the level of concentration and also honesty of the participant. Despite careful instructions, in our experience severely anosmic patients have a tendency to guess the odor instead of stating "I am not able to identify the odor". Patient 1 was markedly anosmic just before the initiation of ADLL-therapy. There is a small increase in her TDI sum-score. But when asked whether she noticed any improvement in smell function over the course of the ADLL therapy she stated NO. In contrast, patient 2 was less impaired – although formally fulfilling the criteria of anosmia. After a few months of ADLL therapy he unsolicitedly stated that he had the impression that his smell function had slightly improved. The TDI scores improved from 10 (before initiation of ADLL-therapy) to 17 under ADLL-therapy and then dropped down again to 13 (anosmic) (still under ADLL-therapy - see Figure 2b in the Main Manuscript). In summary, the long-term therapy with ADLL failed to change the anosmic status of the two IRBD patients. Therefore, we consider it as unjustified to discuss the TDI sum-score changes in patient 2 as a potential sign for an improvement of olfactory function. This issue remains to be studied in RBD patients preferentially with a TDI sum-score of 16 and higher (hyposmia) at baseline. This baseline criterion might increase the likelihood to observe a beneficial effect of ADLL on the non-motor symptom "olfactory impairment".

We have added the above text to Section D) in the Supplementary Material.

In order to notify the explorative nature of the olfactory testing ***the respective paragraph in the Methods section on page 14 reads now as follows***

The TDI sum-score is not agreed as a prodromal progression marker in RBD⁴ and has not been shown so far to be therapy responsive. Therefore, the TDI sum-score is considered as an exploratory measure.

The Result section on page 6 reads now as follows

In respect to the exploration of the olfactory function, the TDI sum-score slightly increased over time with ADLL-therapy, but remained still in the anosmic range for both patients (see also Supplementary Material).

The Discussion section on page 8 reads now as follows

On the other hand, long term therapy with ADLL failed to change the anosmic status of both patients.

Regarding the UPDRS III and MDS-UPDRS III scales, they score action tremor as well as resting tremor separately, however, action tremor (as opposed to resting tremor) is not considered a cardinal motor sign of Parkinson's disease. Therefore, we present UPDRS

III/MDS-UPDRS III scores with resting tremor but without action tremor scores in Figure 2. However, we included the UPDRS III/MDS-UPDRS III scores “with action tremor” in the Supplementary Table S2.

The respective paragraph in the Methods section on page 11 reads now as follows (newly added sentence highlighted in yellow):

“Both patients receiving ADLL-therapy were evaluated with the Unified Parkinson Disease Rating Scale (UPDRS - original version; motor part: UPDRS III – range from 0-104, higher scores indicate greater impairment) or the Movement Disorder Society (MDS)-UPDRS (revised version; motor part: MDS-UPDRS III – range from 0-137, higher scores indicate greater impairment)^{21,22}. Both scores include the item “action tremor”, which is not a cardinal motor sign of PD. We therefore calculated the UPDRS III and the MDS-UPDRS III with and without the item “action tremor” [...].”

The respective paragraph in the Result section on page 6 reads now as follows (newly added sentence highlighted in yellow):

In both patients, regularly assessed UPDRS III/MDS-UPDRS III (motor score – without and with the item “action tremor”) and SCOPA-AUT scores remained normal during ADLL-therapy (Fig. 2a,b, Table S1). Results of the cognitive screening test MoCA stayed in the normal range for patient 2, whereas patient 1 showed a trend of decrease in the MoCA score and developed a mild cognitive impairment during the study. Overall, based on the assessment of the responsible neurologist (AJ) no phenoconversion to Parkinson’s disease or dementia with Lewy bodies [...].”

#Remark 6:

I would suggest Table 1 is converted into 2 Figures (patient 1 then 2) which shows a timeline, below which the scores on various metrics can be shown in comparison to the intervention timing. This would enable a reduction in the number of words describing the imaging timing in the Methods. The details regarding the imaging acquisition could be made more succinct. Most important is the point that the team have use the same SPECT camera to collect the images.

RESPONSE: According to the recommendation of reviewer 1, we have designed a new Figure 2 with part 2a (patient 1) and part 2b (patient 2) based on the values represented in the “old Table 1”, now new Table S1. This figure illustrates the time course of the different clinical and imaging investigations before and under ADLL therapy.

The results of the MoCA, TDI and UPDRS / MDS-UPDRS III tests are shown as line graphs. The time points of the two imaging procedures DAT-SPECT and FDG-PET are depicted by small icons. The data of the imaging procedures are presented in the unchanged former Figure 2a and b, now Figure 3a and 3b.

For completeness of the information, we have shifted the old Table 1 (previously in the Main Document) now into the Supplementary Material as new Table S1.

The respective paragraph in the Results section on page 5/6 reads now (newly added sentence highlighted in yellow):

In Fig. 2 the temporal relationship between the time points of the clinical assessments (motor function test: UPDRS part III or MDS-UPDRS part III; cognitive function test: MOCA; olfactory function test: TDI sum-score) and the time points of the imaging procedures (DAT-SPECT, FDG-PET) is illustrated.

In both patients, regularly assessed UPDRS III/MDS-UPDRS III (motor score – without and with the item “action tremor”) and SCOPA-AUT scores remained normal during ADLL-therapy (Fig. 2a,b, Table S1). Results of the cognitive screening test MoCA stayed in the normal range for patient 2, whereas patient 1 showed a trend of decrease in the MoCA score

and developed a mild cognitive impairment during the study. In respect to the olfactory function a small increase in the TDI sum-score was observed in both patients. However, in summary the long term therapy with ADLL failed to change their anosmic status (see also Supplementary Material). Overall, based on the assessment of the responsible neurologist (AJ) no phenoconversion to Parkinson's disease or dementia with Lewy bodies was observed in both patients [...]"

Finally, we added a new **Figure Legend for Figure 2:**

Figure 2 | Clinical evaluations and imaging procedures of the patients over time. The temporal relationship between the time points of the clinical assessments (motor function test: UPDRS part III or MDS-UPDRS part III; cognitive function test: MOCA; olfactory function test: TDI sum-score) and the time points of the imaging procedures (DAT-SPECT, FDG-PET) is illustrated. The red shaded area indicates therapy with Acetyl-DL-Leucine (5g/day). Results of the clinical ratings are presented for patient 1 (a) and patient 2 (b). Note: during the study the motor function assessment in patient 1 was switched from UPDRS III to MDS-UPDRS III. Motor scores are presented without the item action tremor, which does not belong to the cardinal motor signs of Parkinson's disease. Time points of the imaging procedures are indicated as icons. Results of the imaging procedures are presented in Fig. 3 and Table 1.

#Remark 7:

Table 2 is unnecessary.

RESPONSE: We agree that the Table 2 is overall of low relevance for this manuscript – and in addition it is based on a sequence of non-validated questions in respect to the RBD phenotype.

We have therefore removed Table 2 from the Main Document and have shifted it in the Supplementary Material under the new name Table S2.

See **Page 5 and 15**

(Table S2 in Supplementary Material)

#Remark 8:

Tables 3 and 4 could be combined.

RESPONSE: We have followed the advice of Reviewer 1.

We have combined Table 3 and 4. The combined table is called Table 1. See DISPLAY ITEMS Page 29 of the Main Document.

The title of the new Table 1 was changed to:

Table 1 | Striatal DAT-SPECT binding ratios of patient 1 and patient 2

The legend has been amended accordingly.

#Remark 9:

Page 6 (now page 10) line 134- clarify that the date 2022 refers to the cardiac MIBG scan.

RESPONSE: Thank you for identifying this point of misunderstanding, we have clarified that for patient 2 the date 2022 refers to the cardiac MIBG scan. In addition, we have clarified that patient 1 received the cardiac MIBG scan in 2014.

The respective paragraph in the Methods section on page 10 reads now as follows:

“ [...] and a cardiac ¹²³I-metaiodobenzylguanidine scintigraphy (MIBG) scan^{4,17,18} (patient 1 received the MIBG scan at baseline in 2014, 9 years before treatment with ADLL; patient 2 received the MIBG scan in 2022, under ADLL-treatment) [...]”

#Remark 10:

The drug was withdrawn 10 days before the DATSCAN imaging to prevent any interaction between drug and DATSCAN binding. While the half-life of ADLL in the blood is short, the authors need to discuss what (little) evidence there is that this is sufficient time for the drug to be cleared from the CNS.

RESPONSE: Based on our own unpublished PK data, the plasma terminal half-life ($t_{1/2}$) of the bioactive enantiomer Acetyl-L-Leucine is 0.96 (\pm 0.18) hrs. Therefore, the withdrawal of 10 days is about 240 times $t_{1/2}$ ($>$ 10times of $t_{1/2}$ is assumed to be sufficient). We, however, are not aware of any publication related to the half-life of ALL in brain tissue. Thus, we cannot with certainty exclude the possibility that ALL accumulates in the CNS and thus would still be present in the CNS despite a 10-day withdrawal period. Respective animal experiments are – to our knowledge – not available and must be performed.

We have added the information on plasma terminal half-life into the Main manuscript.

The respective paragraph in the Methods section on page 15 reads now as follows:

“[...] In patient 1, ADLL was withdrawn for 10 days before the DAT-SPECT “under ADLL-therapy” (plasma terminal half-life ($t_{1/2}$) of the active enantiomer Acetyl-L-Leucine is 0.96 (\pm 0.18) hours) in order to avoid any interference of the drug per se with the DAT-SPECT procedure.”

Reviewer #2 (Remarks to the Author):

#Remark 1:

The authors describe an outstanding effect of N-Acetyl-DL-Leucine on Rem Sleep Behavior Disorders of two patients with Parkinson Disease, which was accompanied by increment of putaminal binding ratio of DAT-SCAN SPECT and stabilization of FDG-PET binding (measured according to the ParkinsonDiseaseRelatedPattern-PDRP evaluation). The work is extremely relevant for PD patients.

RESPONSE: We are very grateful for this positive overall evaluation of the manuscript.

#Remark 2:

The works will need additional evidence, as it could be performed in two patients only, but it could provide a starting point for further research.

RESPONSE: We could not agree more.

#Remark 3:

Methodologically the work is sound, clinical and Imaging quantifications are state of art, I could find no flaws in data analysis, the work will be easily reproduceable by other experts.

RESPONSE: Thank you very much for this statement. We agree that our data can be easily reproduced by other experts as the design of the study is simple and feasible in a large number of centers worldwide (self-rating of the severity of the phenotype of RBD and serial DAT-SPECTs).

#Remark 4:

English language is adequate. The authors anticipate the analysis of limitations appropriately.

RESPONSE: We are very grateful for this encouraging statement.

In summary the remarks of Reviewer 2 do not request to change the text in the main document.

REVIEWERS' COMMENTS

Reviewer #1 (Remarks to the Author):

I am happy with the authors' responses to my comments, and congratulate them on this intriguing set of observations.

Reviewer #2 (Remarks to the Author):

I appreciated the manuscript already in the first version, the present version is now expliciting some concepts that could have been unclear to non-experts.